# Uncoupling Protein 1 Does Not Produce Heat without Activation

**DOI:** 10.3390/ijms23052406

**Published:** 2022-02-22

**Authors:** Yongguo Li, Tobias Fromme

**Affiliations:** Chair for Molecular Nutritional Medicine, TUM School of Life Sciences, Technical University of Munich, Gregor-Mendel-Str. 2, 85354 Freising, Germany

**Keywords:** adipocytes, thermogenesis, brown fat, beige/brite cells, molecular brakes, purine nucleotides, lipolysis, feedback mechanisms, obesity

## Abstract

Mitochondrial uncoupling protein 1 (UCP1) is the crucial mechanistic component of heat production in classical brown fat and the newly identified beige or brite fat. Thermogenesis inevitably comes at a high energetic cost and brown fat, ultimately, is an energy-wasting organ. A constrained strategy that minimizes brown fat activity unless obligate will have been favored during natural selection to safeguard metabolic thriftiness. Accordingly, UCP1 is constitutively inhibited and is inherently not leaky without activation. It follows that increasing brown adipocyte number or UCP1 abundance genetically or pharmacologically does not lead to an automatic increase in thermogenesis or subsequent metabolic consequences in the absence of a plausible route of concomitant activation. Despite its apparent obviousness, this tenet is frequently ignored. Consequently, incorrect conclusions are often drawn from increased BAT or brite/beige depot mass, e.g., predicting or causally linking beneficial metabolic effects. Here, we highlight the inherently inactive nature of UCP1, with a particular emphasis on the molecular brakes and releases of UCP1 activation under physiological conditions. These controls of UCP1 activity represent potential targets of therapeutic interventions to unlock constraints and efficiently harness the energy-expending potential of brown fat to prevent and treat obesity and associated metabolic disorders.

## 1. Introduction

Brown and brite/beige adipose tissues are thermogenic and canonically mediate heat generation through mitochondrial respiration that is uncoupled from ATP synthesis via uncoupling protein 1 (UCP1)-mediated proton leak [1,2]. Thermogenesis comes at a high energetic cost. As such, thermogenic adipose tissue functions as a substantial metabolic sink for glucose, fatty acids and amino acids, thereby improving systemic glucose and lipid homeostasis [3]. Beyond its prominent role in energy distribution and expenditure, it is directly implicated in energy intake as a regulator of meal termination and acts as endocrine organ coordinating organismic metabolism [4,5]. Manipulation of BAT activity thus acts on both arms of energy balance, rendering it an attractive target for the treatment of metabolic disease.

In order to appreciate the role of BAT in overall energy expenditure, and thus in the altered energy balance that leads to obesity and associated diseases, it must be realized that BAT is ultimately an energy-wasting organ and thermogenesis comes at a high energetic cost. While this may appear beneficial in the context of widespread metabolic disease today, it most certainly acted as a constraint during evolutionary time spans of food scarcity, rendering UCP1 inherently inactive when not explicitly required and activated and tightly controlled by multiple regulatory layers, including the transcriptional level, mRNA stability and protein degradation, ensuring cell specificity and temporal control. Consequently, simply increasing BAT mass or UCP1 abundance does not lead to an increase in thermogenesis without a concomitant increase or at least maintenance of the sympathetic nervous system (SNS) tone within BAT or alternative activating stimuli [6,7]. Thus, genetic or pharmacological manipulations that promote increases in BAT mass in humans or animal models, in and of themselves, will not guarantee increased thermogenesis and consequent metabolic benefit. Indeed, in accordance with the needs of thermal balance, BAT thermogenesis is actively switched off under a variety of circumstances including the thermoneutral conditions modern humans live in most of the time. Recruitment of UCP1 abundance is therefore a concept entirely different and distinct from UCP1 activity.

It is clear from this derivation that to efficiently harness the energy-converting potential of BAT for the prevention and treatment of obesity and diabetes requires not only knowledge of the substrates and metabolic pathways involved, of the control of these pathways by neurotransmitters and hormones and of the control of gene expression in BAT, but also an appreciation and understanding of the inherently inactive nature of UCP1, the physiological and molecular brakes of UCP1 activation and the exquisite sensitivity of UCP1 activity to temperature-dependent regulation. In this review, we highlight the inherently inactive nature of UCP1, with a particular emphasis on the molecular brakes of UCP1 activation under physiological conditions. We propose that these brakes on UCP1 activity represent potential targets of therapeutic interventions to unlock constraints and efficiently and fully harness the energy-converting potential of brown fat to prevent and treat obesity and associated metabolic disorders.

## 2. UCP1 Is Not Leaky

Thermogenesis in brown fat is under tight, unconscious sympathetic control and inherently inactive at the molecular level. The crucial hub of regulation is formed by the functional core of the thermogenic machinery itself, mitochondrial UCP1 and its molecular activity. When active, UCP1 mediates a net proton translocation from the intermembrane space into the mitochondrial matrix, thus diminishing proton motive force in the absence of ADP phosphorylation. The exact molecular mechanism is still under debate [8]. Accumulating evidence points to a functional role of free fatty acids acting both as transport substrate for the solute carrier UCP1 and as proton shuttle via its terminal carboxyl group [9]. Be it by this or another physical interaction, free fatty acids act as the natural activators of UCP1 uncoupling function, being liberated lipolytically as a result of sympathetic, adrenergic signaling [10]. In the absence of free fatty acids, UCP1 is constitutively inhibited by physical binding of the abundant di- and triphosphate forms of purine nucleotides (ADP, ATP, GDP, GTP) [11].

It has long been debated whether UCP1 contributes to basal proton conductance of brown fat mitochondria in the absence of adrenergic or other lipolytic stimuli [12,13]. This question of basal leak activity is of far more than academic interest. In the case of UCP1 possessing a basal proton conductance, merely increasing the amount of UCP1 must be expected to contribute to proton leakage and consequently energy expenditure, possibly exploitable to provoke fat mass loss. If, however, UCP1 does not possess such a residual proton conductance, therapeutic solutions with the combined goal of not only recruiting but also of activating brown adipose tissue/UCP1 are mandatory.

Notably, comparing UCP1-deficient mitochondria and wildtype controls, no difference in basal proton conductance was observed measuring the overall proton leak kinetics [14]. Moreover, the basal respiration of cultured primary brown cells is identical regardless of whether they possess UCP1 [13,15]. This demonstrates that UCP1 does not show any proton transport activity when non-stimulated, that is to say, it is not ‘leaky’ per se [13] (i.e., it does not allow for proton flux over the mitochondrial membrane when it is not directly stimulated). Consistently, ectopically expressed UCP1 does not contribute to basal proton conductance of human cultured cells, emphasizing the requirement of UCP1 activation for boosting energy expenditure [16]. 

Lastly, there are no differences in basal metabolic rate between mice with and without UCP1 [17,18]. This further confirms the tenet that UCP1 is not leaky and does not contribute to basal metabolic rates. To quote the phrase that “UCP1 mRNA does not produce heat” [19], we here advocate that neither does unactivated UCP1 protein produce heat. 

## 3. Activated UCP1 Protein Produces Heat

Brown adipose tissue and mitochondrial UCP1 residing therein needs to be activated to produce heat, namely, by free fatty acids liberated through lipolysis. Within minutes of cold exposure, thermal afferent signals elicited by thermoreceptors in the skin are transmitted to the brain, excite the sympathetic nervous system, stimulate sympathetic outflow to BAT and noradrenaline secreted by neuronal varicosities interacts with adrenergic receptors on brown adipocytes. Intracellularly, lipolysis is induced through the canonical adrenergic receptor–Gs-protein–adenylyl cyclase–cAMP–protein kinase A (PKA) pathway. Free fatty acids released from lipid droplets act as both fuel for mitochondrial β-oxidation and activators of UCP1 (reviewed in [1]). Due to their essential role, utilization of extracellular sources of fatty acids has evolved to compensate a possible absence [20] or intracellular inability to mobilize [21,22] internal fatty acid stores.

The resultant physical, UCP1-dependent heat production is uncontroversial and has been demonstrated by a score of different methods in as many biological model systems: 

Stimulated heat production of brown fat mitochondria has been measured by microcalorimetry [23] and similarly in cultured and isolated brown adipocytes [24], leading to inducible temperature increases as measured by infrared thermography [25] and intracellular fluorescent sensors [26,27]. Alternatively, heat production can be calculated from the caloric equivalent of oxygen consumed. Such respirometric experiments unequivocally demonstrate the requirement of UCP1 for adrenergically induced heat production in brown adipocytes [15,28], supported by a similar UCP1-dependency in induced substrate fluxes [29].

In living mice, brown adipose tissue produces heat in response to cold exposure and adrenergic stimulation, as measured by implanted temperature sensors [5,30]. This effect quantitatively depends on the presence of UCP1, as demonstrated by indirect calorimetry [31] and infrared thermography [32]. Similarly, cold exposure increases human brown fat-specific energy expenditure and oxygen consumption [33,34] and glucose uptake [35,36], as determined by positron emission tomography in combination with computer tomography. Skin surface temperature measured by infrared thermography accordingly increases upon cold exposure in these exact anatomical locations [37].

Taken together, while neither UCP1 mRNA nor unactivated protein produces any heat, activated UCP1 (only) unequivocally does precisely that, as is verifiable at all levels from isolated mitochondria and cultured cells to brown adipose tissue depots in mice and men.

## 4. Recruitment of UCP1 Is Distinct from Activation

The best-studied scenario of BAT recruitment is undoubtedly the most obvious one, namely, cold exposure. In this condition, activation of thermogenesis and recruitment of thermogenic capacity occurs as a consequence of the same signal: sympathetic stimulation, i.e., norepinephrine release within the tissue [38]. Importantly, activation and recruitment are not obligatorily synchronous and several physiological situations coerce decoupling. In precocial newborns, for instance, BAT is present and fully functional with a considerable thermogenic capacity at birth already [39]. A chronically high sympathetic tone driving both BAT recruitment and constant activity during intrauterine life at 37 °C appears implausible, indicating the existence of an alternative, potent and fully competent non-adrenergic recruiting mechanism. Seasonal preparation for hibernation serves as a further example, a recruitment process where the adrenergic pathway would be contraindicated, since during the time when BAT is recruited, i.e., in warm late summer, hibernators accumulate fat reserves for the winter. Accordingly, recruitment of brown fat in hibernators is non-adrenergic [39,40]. Thus, induction of UCP1 expression and full recruitment of BAT capacity can occur in response to alternative adrenergic and non-adrenergic signals.

One such alternative, potent and fully competent mechanism for BAT recruitment is the PPARγ pathway, probably responsible for the recruitment during pre-hibernation and in the prenatal state. In fact, continuous treatment with the PPARγ agonist rosiglitazone robustly increases the amount of thermogenically competent, UCP1-expressing adipocytes, which are capable of responding to adrenergic stimulation with a dramatic increase in UCP1-mediated oxygen consumption [15,39]. Despite this compelling evidence from cellular model systems, the in vivo relevance remains to be validated, as is the nature and source of the still enigmatic, natural, endogenous PPARγ ligand [39].

In summary, while doubt remains as to the mechanisms, there clearly are physiological conditions in which brown fat capacity needs to be recruited but, importantly, *not* thermogenically activated at the same time. Figuratively, BAT recruited in this way represents a larger stove waiting for the spark, an adrenergic stimulus, to initiate heat production. The same rationale must certainly apply to pharmacological or genetic or any other artificial increase in BAT mass or UCP1 abundance. 

## 5. Controlled Activation of Brown Fat Can Reveal Maximal Thermogenic Capacity, Not Actual Thermogenic State

In the preceding parts, we argued that the amount of UCP1 in brown adipose tissue depots—natural or manipulated—is not a measure of thermogenic activity due to the inherently inactive nature of UCP1 in the absence of an adrenergic stimulus. An obvious conclusion then appears to activate brown fat by norepinephrine or synthetic, adrenergic agonists to alleviate this concern [41]. This certainly is a more informative way to demonstrate physiological consequences of altered BAT or UCP1 mass. Conceptually, such an experiment will, however, still be a measure of capacity, not endogenous activity (Figure 1). 

In cultured brown adipocytes, for instance, UCP1 abundance will not increase baseline activation (see above). At the same time, stimulated full activation does also not allow additional conclusions beyond the functional capacity of the cell model. In vivo, accordingly, the classical norepinephrine test—injection of norepinephrine/CL-316243 during indirect calorimetry—will only allow the quantification of thermogenic capacity, not the contribution of any amount of brown fat or UCP1 to metabolism under free-running conditions.

Acute thermogenic power is necessarily tightly controlled and subject to feedback mechanisms. In essence, the amount of heat produced is regulated to match the amount of heat lost, to preserve normothermia. A very vivid example is the production of extra heat by treating animals with the chemical uncoupler DNP. This intervention leads to a robust increase in metabolic rate at thermoneutrality but fails to do so at colder ambient temperature [42] because the increase in energy expenditure generated by DNP-mediated uncoupling is compensated by feedback reducing BAT thermogenesis. The same is true for peripheral hyperthyroidism, where thyroid hormone induces hyperthermia and leaves thermogenic brown and beige fat metabolically inactive as additional UCP1-mediated thermogenesis is not required for maintaining body temperature [43]. Along the same line, the same absolute amount of heat will be generated in response to a certain ambient temperature in both cold- and warm-acclimated animals, despite large differences in brown fat thermogenic capacity and employed mechanisms [17,18]. Thus, even cold exposure as the natural activator of brown adipose tissue will not reveal consequences of altered UCP1 expression beyond thermogenic limit capacity.

One way to circumvent this conundrum is to directly monitor brown fat thermogenesis in real-time using thermosensors that are implanted in proximity to interscapular BAT under free-running conditions [5]. This physiological readout is unbiased by the animal activity as observed by indirect calorimetry or infrared thermal imaging [44]. However, it will be essential to assess the validity of BAT temperature measurements using UCP1 WT and KO mice to control for the cofounding influence by changes in core body temperature [31]. 

Collectively, physiological changes in response to a manipulated UCP1 abundance cannot be causally attributed to endogenous, UCP1-dependent thermogenesis, neither by indirect calorimetry in vivo nor by cellular respirometry, neither in a basal state nor activated by cold or norepinephrine.

## 6. Pharmacological but Not Physiological Activation of UCP1 Protects against Diet-Induced Obesity

We are unable to determine physiological consequences of altered UCP1 *abundance* beyond thermogenic capacity by activating thermogenesis. However, one can certainly determine such consequences of altered UCP1 *activity*.

Thermogenesis inevitably comes at a high energetic cost. The full activation of UCP1-mediated energy-dissipating mechanisms can contribute to whole body energy substrate homeostasis, exert significant metabolic benefits and protect from a chronic imbalance between energy intake and expenditure. This concept emerged in the late 1970s and resulted from two key observations [45]: (1) feeding an obesogenic diet induces a recruitment of BAT [46]; (2) decreased BAT activity in obese (ob/ob) mice [47,48]. These pioneering observations established a link between BAT and energy balance, and resulted in a new paradigm in the etiology of obesity [45]. Impairment of the thermogenic activity of BAT is subsequently regarded as a causative factor in the development of obesity in a number of obese animal models [49,50,51]. This concept appeared to be corroborated by UCP1-KO mice developing obesity at thermoneutrality [52], an observation as controversial as its apparent implication. 

Nevertheless, there is considerable controversy regarding the contribution of brown fat to prevent the development of diet-induced obesity, even though a recruitment of UCP1 in response to HFD feeding has often been reported [53]. The most important and stringent model for robust validation of the metabolic functions of UCP1 is the UCP1 knockout (KO) mouse. This model has varyingly been reported to be either protected from diet-induced obesity [52,54,55,56,57] or not [6,58,59,60,61,62,63] at thermoneutral conditions. The reason for this confusing pattern of varying outcome—even between cohorts of the same colony [64]—is currently unknown. Based on the combined evidence, clearly, the presence and recruitment of UCP1 in BAT during HFD does not have a robust, if any, effect on the development of body mass.

In summary, UCP1 *per se* does not protect mice from diet-induced obesity under physiological conditions, irrespective of the type of diet or the ambient temperature. Pharmacological activation of UCP1, however, by daily injection of the selective β3-adrenergic receptor agonist CL316,243, resulted in a significant reduction of body weight in a UCP1-dependent manner [6,65]. In fact, the anti-obesity effect of β3-AR stimulation has been well documented in various animal models of obesity [66]. Therefore, UCP1 fails to protect against diet-induced obesity due to insufficient activation. Possibly the threshold for recruitment of UCP1 versus activation of UCP1 is different under HFD feeding or the activation of UCP1 is impaired during a HFD challenge. 

In humans, the anti-obesity and metabolic efficacy of the β3-adrenergic agonist CL316243 is limited for multiple reasons, such as low efficacy at the human β3AR, low β3AR expression levels, a limited amount of brown fat and cardiovascular side-effects. The β3-specific agonist mirabegron (BETMIGA^®^) caused enhanced energy expenditure in humans (+203 ± 40 kcal/day) [67]. However, it is not clear whether the mirabegron doses used also activate β1 [68] or β2 adrenoceptors [69]. Moreover, treatment with this drug resulted in significant cardiovascular side effects (hypertension and tachycardia) [69]. Thus, alternative and novel activators of brown and beige fat cells have to be identified. 

## 7. Physiological and Molecular Brakes of Brown Fat Activation and Their Release

The inherently inactive nature of UCP1 impedes quantification of its contribution to energy balance and complicates the pharmacological targeting of UCP1 abundance and brown fat cell number. On the other hand, negative regulators of thermogenic activity may serve as a largely unexplored and unexploited resource of novel target mechanisms.

In the broadest physiological sense, brown fat activity is negatively regulated by any environmental, allometric or behavioral condition that causes reduced thermogenic demand (Figure 2). This includes, most obviously, ambient temperature that linearly scales with thermogenic intensity between maximal metabolic rate and the lower critical temperature of the thermoneutral zone or, in simple words, in mild to intense cold [41,70]. Beyond ambient temperature, defended body temperature exerts a profound impact on brown fat activity, which is suppressed during hibernation and deep torpor [71,72]. Additionally, extra heat production imposed by exercise, pregnancy and lactation replaces, and thus suppresses, BAT thermogenesis (see review [70]). 

At a given temperature gradient, heat flow is a function of insulation and interface surface. The former is a highly relevant determinant of thermogenic demand, as demonstrated in mouse models of furlessness and skin barrier defects [73,74,75]. The latter, surface-to-volume ratio, is an allometric function of body mass. Accordingly, smaller animals, including human infants, are more dependent on BAT thermogenesis than bigger animals [76,77] and both relative mass and capacity of brown fat are inversely correlated with body mass [78]. A change in surface-to-volume ratio is also the underlying mechanism of social thermoregulation, i.e., huddling, an efficient cooperative behavior to save energy [79]. Indeed, increased housing density of mice per cage (even two mice) suppresses the thermogenic capacity of brown adipose tissue compared to single housing [80,81]. This may be one of the factors underlying variability in response to HFD. 

Finally, thermogenesis comes at a high energetic cost. When there is insufficient energy available, as in response to fasting, thermogenesis is severely impaired to conserve energy. The fasting-induced reduction in thermogenic capacity is linked to thyroid hormone levels [82] and BAT atrophy [83]. In short, metabolic fuel availability provides permissive signals to BAT thermogenesis.

Together, these phenomena form another major hurdle for brown fat-based therapeutic strategies, beyond the limited amount of human brown fat. Modern humans spend much of their time at thermoneutrality, possibly explaining a low brown fat amount and capacity in humans. Furthermore, patients benefiting most from a novel treatment for metabolic disease will typically be of larger than average body mass. In the following, we review the neural, endocrine and molecular mechanisms underlying suppression of BAT activity. All of them constitute potential constraints on future BAT-centric therapies and at the same time, and for the same reason, promising targets for complementary intervention.

### 7.1. Nerve and Vascular Rarefaction

BAT is highly innervated by the sympathetic nervous system and contains a rich vascular supply [70]. Both characteristics are essential functional components of its thermogenic function. While the sympathetic nervous system (SNS) controls BAT thermogenesis through the release of norepinephrine (NE) and β-adrenergic signaling [84], the vasculature supplies oxygen and nutrients to the tissue and distribution of heat to the other parts of the body [85]. Therefore, the status of innervation and vasculature within the BAT microenvironment is critical for its function both in terms of thermogenesis and systemic metabolic homeostasis [86,87,88,89,90,91,92]. Chemical and surgical denervation studies have clearly demonstrated that adipose tissue neural innervation is crucial for maintaining proper metabolic health. Capillary rarefaction leads to hypoxia and inflammation, which consequently reduces β-adrenergic signaling and leads to ‘whitening’ of BAT [93]. Thus, risk factors that are associated with nerve and vascular rarefaction contribute to the impairment of BAT function. In fact, peripheral neuropathy and capillary rarefaction are the most frequent neurological complication in diabetic patients [94]. Analysis of BAT and WAT in obesogenic mouse models have revealed dramatic reductions in sympathetic innervation [95]. It is reasonable to suspect that HFD-induced nerve and vascular rarefaction functions in a vicious circle that might contribute to the “insufficient” activation of brown fat.

### 7.2. Norepinephrine Clearance

Sympathetic tone is determined by two key parameters: norepinephrine (NE) release, a direct function of sympathetic activity, and extracellular NE clearance, which is key to terminating adrenergic signaling. Historically, NE clearance was believed to occur exclusively through active transport via the monoamine transporter SLC6A2 of SNS nerve terminals [96]. After cellular reuptake, NE can either be degraded by mitochondrial monoamine oxidase (MAO) or sequestered in NE storage vesicles to be recycled. However, recent studies demonstrated that proinflammatory sympathetic neuron-associated macrophages in adipose tissue [97,98,99,100], as well as adipocytes [101], can take up and degrade catecholamines released from sympathetic neurons through SLC6A2 and organic cation transporter 3 (Oct3), respectively, functioning as additional clearance mechanisms to clear catecholamines in fat tissue microenvironments. Obviously, multiple pathways act in concert to safeguard metabolic thriftiness. 

### 7.3. Cyclic Nucleotide Generation and Degradation

A well-established paradigm for brown fat activation is activation of Gs protein-coupled receptors (GPCRs) initiating cAMP generation [10]. Vice versa, agonist binding to Gi-coupled GPCRs results in an inhibition of adenylyl cyclase followed by a decrease in cAMP, inactivation of PKA and, finally, dephosphorylation of lipolytic machinery. Consequently, lipolysis is inhibited and thermogenesis is terminated. Within this concept, ligands of Gi-coupled receptors are important autocrine/paracrine molecular brakes of brown fat activation and are able to fine-tune heat production to the thermogenic demand (for a review, see [10] and [102]). 

The second messengers cAMP and cGMP are central for thermogenic activation of brown adipocytes [103]. Phosphodiesterases (PDEs) are built-in enzymes that hydrolyze cAMP and cGMP. Therefore, activation of PDEs increases degradation of cAMP and thus attenuates PKA activity, resulting in net dephosphorylation of HSL and dampening of lipolysis. Of note, stimulation with the β-adrenergic agonist isoproterenol results in both increased cAMP synthesis and increased cAMP degradation through phosphorylation of PDEs in adipocytes, mostly PDE3B and PDE4 [103]. Consistent with a feedback regulatory mechanism, this is believed to contribute to the fine-tuning of cAMP levels and PKA activity and thereby control of lipolysis and thermogenesis [102].

### 7.4. PKA Regulatory Subunits

PKA-mediated lipolysis activation represents the canonical pathway to stimulate thermogenesis. When inactive, PKA is a tetrameric holoenzyme comprising two regulatory (R) subunits and two catalytic (C) subunits [103]. Binding of the C-subunit to the inhibitory sites of the respective PKA regulatory subunits renders the kinase inactive. Elevated cAMP levels promote the dissociation of PKA holoenzymes and the release of catalytic subunits (PKA-C). Once unleashed, free active PKA-C subunits are competent to phosphorylate their target proteins, e.g., key lipolytic proteins, and initiate lipolysis. Lipolysis-derived free fatty acids (FFAs) act as both fuel for mitochondrial β-oxidation and activators of UCP1. The mechanism of PKA-C activation and signaling has been largely characterized [104]. How exactly rapid recapture of liberated catalytic subunits happens, however, has been a longstanding unresolved question. In any case, re-association of liberated PKA-C to inhibitory PKA-R subunits serves as a termination mechanism. Consistently, targeted disruption of the RIIβ subunit of protein kinase A in mice leads to chronic activation of brown adipose tissue, increased metabolic rate, elevated body temperature and an UCP1-independent lean phenotype, supporting a critical role for RIIβ in restraining PKA activity [105]. Lastly, even though multiple subunits of PKA, including RII beta, R1 alpha, C alpha and C beta 1, are expressed in fat cells, regulatory subunits are expressed much more highly than catalytic subunits to support rapid catalytic subunit re-association [106]. Altogether, PKA regulatory subunits participate in keeping PKA activity at bay and facilitate a fast switch-off of PKA-dependent cellular responses, including thermogenesis.

### 7.5. Protein Phosphatases

Ultimately, cessation of lipolysis depends on the dephosphorylation of PKA targets, such as HSL and perilipin, by protein phosphatases; therefore, cAMP degradation and PKA inactivation do not per se terminate the lipolytic cascade and thermogenesis [102]. Through protein phosphatase (PP)-mediated dephosphorylation of phosphoproteins, protein kinase activity is counteracted to reach a delicate balance, thus ensuring high signal fidelity. Although activation of lipolysis by PKA has been well studied, inactivation via protein phosphatases is poorly understood. Adipocyte homogenates contain approximately equal activities of PP1 and PP2A, lower levels of PP2C and virtually no PP2B activity [107]. However, the specific phosphatase responsible for the dephosphorylation of lipolysis-related proteins HSL and perilipin remains elusive. Of note, loss of phospholipase C-related catalytically inactive protein (PRIP), a binding partner of PP1 and PP2A, triggers hyperphosphorylation of HSL and perilipin, activation of UCP1 and exhibits a lean phenotype with a reduced amount of white adipose tissue [108]. Essentially, PRIP promotes the translocation of phosphatases to the surface of lipid droplets to trigger the dephosphorylation of HSL and perilipin, thus reducing PKA-mediated lipolysis and thermogenesis. Therefore, protein phosphatase-mediated dephosphorylation of lipolytic machinery is the most crucial and ultimate step for terminating lipolysis. 

### 7.6. Nucleotides

UCP1, the central mechanistic component of brown fat-mediated non-shivering thermogenesis, is itself a regulatory center inhibited by purine nucleotides (GDP, GTP, ADP, ATP) [109]. Purine nucleotides act as a default shut-off mechanism of UCP1 in the absence of thermogenic demand [11,110]. Upon BAT activation by the sympathetic nervous system, lipolysis-derived free fatty acids are thought to act as thermogenic activators via displacing inhibitory purine nucleotides from UCP1 in a competitive manner. Nevertheless, the nature of purine nucleotide interaction with fatty acids remains unresolved. Cytosolic free purine nucleotide concentrations are adrenergically modified by several routes acting in concert, including calcium complex formation and enzymatic nucleotide degradation. Moreover, the inhibitory effect of purine nucleotides is pH dependent: strong at low and weak at high pH values, respectively [111]. Upon activation of thermogenesis in a brown fat cell, cytosolic pH increases to up to 8.0 [112]. Cytosolic alkalizationwill thus contribute to the release of bound nucleotides and facilitate UCP1 activation by FFA. Therefore, adrenergic activation of UCP1-mediated non-shivering thermogenesis includes the concerted action of free fatty acid liberation, remodeling of nucleotide pool size by enzymatic degradation and the fraction of inhibitory free nucleotides by changes in calcium concentration and pH [110]. Reducing the amounts of nucleotides binding to UCP1 presents itself as an alternative target to modulate UCP1 activity.

## 8. Conclusions

Our understanding of the physiological functions of BAT has continuously evolved. It is now evident that it fulfils multiple important roles beyond the core of thermoregulation. The energy-expending and metabolic sink properties of activated BAT make it an attractive target for combating obesity and metabolic disease. However, brown fat is ultimately an energy-wasting organ and under tight control. Consequently, brown fat thermogenesis is highly sensitive to changes in ambient temperature and is only activated by heat loss, less active when temperature rises and even repressed when ambient temperature reaches the thermoneutral zone. Since humans usually live in an environment close to thermoneutrality, this represents a major hurdle for the development of brown fat-based therapeutic strategies against metabolic disorders, regardless of whether BAT is present in sufficient amounts. In short, due to built-in brake systems, UCP1 is inherently inactive and does not produce heat without activation. Therefore, understanding the brake systems on brown fat thermogenesis may reveal novel targets of therapeutic intervention to unlock constraints and efficiently harness the energy-expending potential of brown fat.

## Figures and Tables

**Figure 1 ijms-23-02406-f001:**
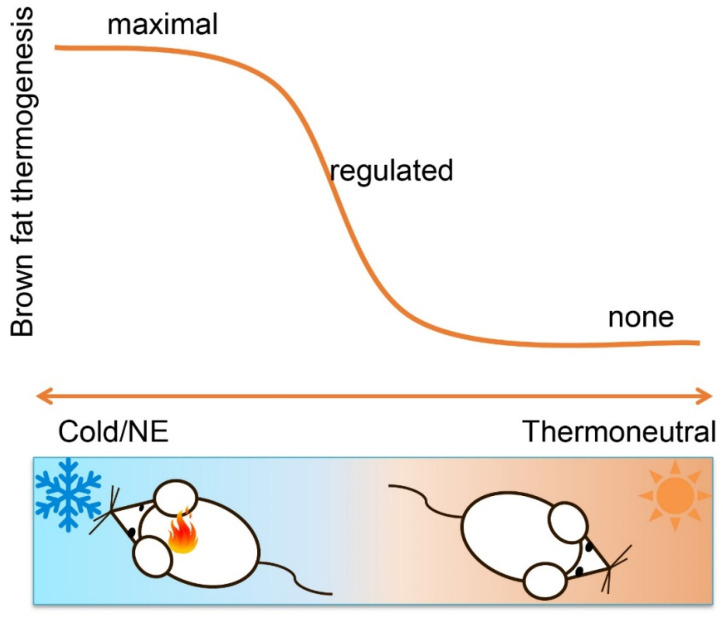
Brown fat thermogenesis is dynamic, fine-tuned and highly sensitive to changes in ambient temperature. It can be fully activated by cold exposure or norepinephrine treatment, is less active when temperature rises and even repressed when ambient temperature reaches the themoneutral zone. As such, activation of brown fat by adrenergic activation can reveal maximal thermogenic capacity, not actual thermogenic state. Moreover, actual thermogenesis is tightly regulated and subject to feedback regulation. Non-brown fat-mediated heat production, such as synthetic uncoupler and peripheral hyperthyroidism, can leave thermogenic brown and beige fat metabolically reduced or even inactive.

**Figure 2 ijms-23-02406-f002:**
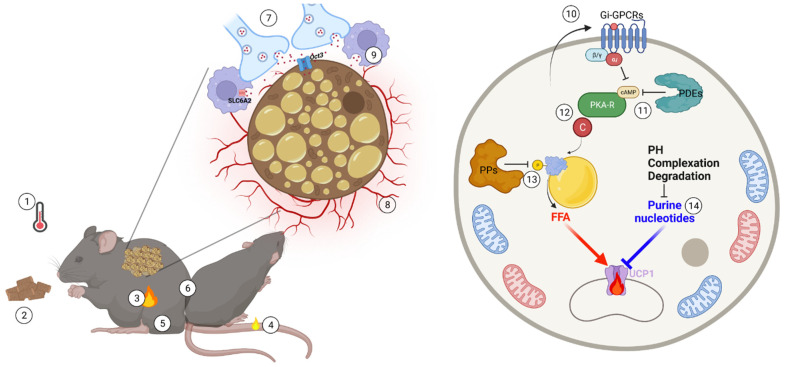
Physiological and molecular brakes of brown fat activation and their release. In accordance with the needs of thermal balance, brown fat thermogenic power is necessarily tightly controlled and subject to synergistic feedback mechanisms. Thermogenesis is highly sensitive to many physiological factors, such as environmental temperature (1), energy state (2), endogenous heat production (3), heat loss (4), body mass (5) as well as social behavior, such as huddling (6). At the tissue and molecular level, the complex thermogenic microenvironment and molecular networks constraining brown fat activity include nerve (7) and vascular (8) rarefaction, norepinephrine clearance (9), paracrine/endocrine Gi-mediated GPCR signaling (10), intracellular cAMP degradation (11), PKA activity fine-tuning (12), phosphatase (PP)-mediated dephosphorylation of phosphoprotein (13) and purine nucleotide metabolism remodeling (14). These built-in brake systems efficiently titrate the local adaptation required to accurately meet constantly varying thermogenic demands.

## Data Availability

Not applicable.

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
