# Peer review of "Uncoupling Protein 1 Does Not Produce Heat without Activation"

_ijms, 2022, doi:10.3390/ijms23052406_

Round 1

Reviewer 1 Report

This is an interesting summary of the role of UCP1 activation in thermogenesis. This manuscript is a very informative review. In particular, the assertion that the UCP1 activation is required for thermogenesis is a departure from current studies that focus only on the expression level of UCP1. In this sense, this manuscript is in a timely manner.

By the way, in a previous paper (Nutr Food Res, 2021, 65(2), e2000681), the author reported that food ingredients that have been reported to enhance thermogenesis by increasing UCP1 expression lost their effect under thermoneutral conditions. Could this be due to the fact that UCP1 activation does not occur under thermoneutral conditions due to the low heat production required, as described here? If this is a possibility, it would be good to mention it in this manuscript now that the increase in thermogenesis by food components is attracting attention.

Author Response

This is an interesting summary of the role of UCP1 activation in thermogenesis. This manuscript is a very informative review. In particular, the assertion that the UCP1 activation is required for thermogenesis is a departure from current studies that focus only on the expression level of UCP1. In this sense, this manuscript is in a timely manner.

We thank the reviewer for favoring our manuscript.

By the way, in a previous paper (Nutr Food Res, 2021, 65(2), e2000681), the author reported that food ingredients that have been reported to enhance thermogenesis by increasing UCP1 expression lost their effect under thermoneutral conditions. Could this be due to the fact that UCP1 activation does not occur under thermoneutral conditions due to the low heat production required, as described here? If this is a possibility, it would be good to mention it in this manuscript now that the increase in thermogenesis by food components is attracting attention.

We thank the reviewer for pointing out our previous paper. In fact, we found that dietary fish oil supplementation failed to recruit UCP1 expression at thermoneutral conditions, neither alone nor in synergy with the β3-adrenergic route of thermogenesis. Therefore, in our opinion, the null-effect phenotype is not due to low thermogenesis demand.

Reviewer 2 Report

Summary

In this review article, Li et al. nicely overviews the current knowledge around the molecular regulation of UCP1-activation. This article conveys a clear message reminding people in the field that UCP1 expression does not indicate UCP1 function, and that there exists a important layer of regulation for UCP1 functional activation. Particularly, interpretation from the evolutionary point of view was excellent. This is important in view of the current trend in studying the brown/beige fat recruitment as a therapeutic approach against obesity. The review covers various topics from mutant mouse models, experimental approaches for investigating the activation, and molecular players negatively regulating UCP1.

With only a minor points, I would highly recommend this article for publication.

Minor comments

  • UCP1 appears to be tightly regulated at the transcriptional level, as well as the mRNA stability degradation, ensuring cell-specificity (limiting expression) and temporal control. It is very interesting that the system has evolved to place many layers of regulation to control UCP1 expression and activity, consistent with the authors’ implication that activated thermogenesis itself is an energy wasting process that should be avoided especially when food scarcity is the norm during evolution. The authors may comment on this aspect of UCP1 regulation at the expression levels in the background, if space is allowed.
  • It has been reported that fatty acids from BAT itself is dispensable using Dgat knockout mouse (Chitraju et al. 2020, Cell Reports). The authors should comment on this work, when describing FFA from lipolysis are fuels for UCP1. (Section 3, line110)
  • Analysis of the innervation of BAT (and WAT) under obesogenic mouse models have been reported, demonstrating reduction of sympathetic innervation. (Wang and Loh et al., 2020, Nature). The authors can cite this paper at line 243/244 when describing the innervation patterns under obesogenic conditions.

Author Response

In this review article, Li et al. nicely overviews the current knowledge around the molecular regulation of UCP1-activation. This article conveys a clear message reminding people in the field that UCP1 expression does not indicate UCP1 function, and that there exists a important layer of regulation for UCP1 functional activation. Particularly, interpretation from the evolutionary point of view was excellent. This is important in view of the current trend in studying the brown/beige fat recruitment as a therapeutic approach against obesity. The review covers various topics from mutant mouse models, experimental approaches for investigating the activation, and molecular players negatively regulating UCP1.

With only a minor points, I would highly recommend this article for publication.

 We appreciate the reviewer for favoring our manuscript.

Minor comments

  • UCP1 appears to be tightly regulated at the transcriptional level, as well as the mRNA stability degradation, ensuring cell-specificity (limiting expression) and temporal control. It is very interesting that the system has evolved to place many layers of regulation to control UCP1 expression and activity, consistent with the authors’ implication that activated thermogenesis itself is an energy wasting process that should be avoided especially when food scarcity is the norm during evolution. The authors may comment on this aspect of UCP1 regulation at the expression levels in the background, if space is allowed.

We thank the reviewer for this excellent suggestion. We have added this aspect of tight UCP1 regulation. See line 44-46.

  • It has been reported that fatty acids from BAT itself is dispensable using Dgat knockout mouse (Chitraju et al. 2020, Cell Reports). The authors should comment on this work, when describing FFA from lipolysis are fuels for UCP1. (Section 3, line110)

We thank the reviewer for this suggestion. We have included this work in line 113-116.

  • Analysis of the innervation of BAT (and WAT) under obesogenic mouse models have been reported, demonstrating reduction of sympathetic innervation. (Wang and Loh et al., 2020, Nature). The authors can cite this paper at line 243/244 when describing the innervation patterns under obesogenic conditions.

We thank the reviewer for this suggestion. We have included this work in line 312-313 as well.